# The Dietary Intake and Practices of Adolescent Girls in Low- and Middle-Income Countries: A Systematic Review

**DOI:** 10.3390/nu10121978

**Published:** 2018-12-14

**Authors:** Emily C. Keats, Aviva I. Rappaport, Shailja Shah, Christina Oh, Reena Jain, Zulfiqar A. Bhutta

**Affiliations:** 1Peter Gilgan Centre for Research and Learning, Centre for Global Child Health, The Hospital for Sick Children, Toronto, ON M5G 0A4, Canada; emily.keats@sickkids.ca (E.C.K.); rappaportaviva@gmail.com (A.I.R.); shailja.shah@sickkids.ca (S.S.); christina.oh@sickkids.ca (C.O.); reena.jain@sickkids.ca (R.J.); 2Department of Paediatrics and Child Health, Division of Woman and Child Health, Aga Khan University, Karachi 74800, Pakistan

**Keywords:** diet, adolescent girls, developing countries, dietary practices, energy intake, nutrition transition

## Abstract

In many low- and middle-income countries (LMICs) the double burden of malnutrition is high among adolescent girls, leading to poor health outcomes for the adolescent herself and sustained intergenerational effects. This underpins the importance of adequate dietary intake during this period of rapid biological development. The aim of this systematic review was to summarize the current dietary intake and practices among adolescent girls (10–19 years) in LMICs. We searched relevant databases and grey literature using MeSH terms and keywords. After applying specified inclusion and exclusion criteria, 227 articles were selected for data extraction, synthesis, and quality assessment. Of the included studies, 59% were conducted in urban populations, 78% in school settings, and dietary measures and indicators were inconsistent. Mean energy intake was lower in rural settings (1621 ± 312 kcal/day) compared to urban settings (1906 ± 507 kcal/day). Self-reported daily consumption of nutritious foods was low; on average, 16% of girls consumed dairy, 46% consumed meats, 44% consumed fruits, and 37% consumed vegetables. In contrast, energy-dense and nutrient-poor foods, like sweet snacks, salty snacks, fast foods, and sugar-sweetened beverages, were consumed four to six times per week by an average of 63%, 78%, 23%, and 49% of adolescent girls, respectively. 40% of adolescent girls reported skipping breakfast. Along with highlighting the poor dietary habits of adolescent girls in LMIC, this review emphasizes the need for consistently measured and standardized indicators, and dietary intake data that are nationally representative.

## 1. Introduction

The adolescent cohort (aged 10–19 years) today (1.2 billion) represents the largest generation in history [1], and 90% of these adolescents reside in a low- or middle-income country (LMIC) [2]. While some populations have stabilized, projections estimate a 42% growth of youth (15–24 years) in Africa alone from 2015 to 2030 [3], underpinning the importance of this group, and especially girls and young women, in driving global health and development. Undoubtedly, reaching adolescents will be critical to achieve each of the Sustainable Development Goal targets, but particularly those relating to health, poverty, education, and the reduction of inequalities [4].

Adolescence is a unique period that represents immense biological and socio-emotional development, typically alongside increased autonomy. The benefits of encouraging healthy practices in adolescence are likely to extend into adulthood and throughout the life course, though, until recently, the health and wellbeing of adolescents has been largely overlooked in policy and programming [4].

We know that the nutrient needs of this group are high and, in LMICs specifically, adolescents are at risk of malnutrition (underweight, overweight, micronutrient deficiencies) [5,6,7,8]. Deficiencies of iron disproportionally affect adolescent girls and can affect more than 30% of girls in countries with lower social development index (SDI) scores [9]. Vitamin A deficiency has a prevalence of 20% among girls aged 10–14 years and 18% among girls aged 15–19 years in low SDI countries, whereas iodine deficiency affects 3% and 5% of younger and older girls, respectively [9]. In 2016, the global prevalence of underweight was 8%, while overweight and obesity reached over 5% of young girls [9]. In many LMICs, the rates of overweight and obesity now surpass that of underweight [5,6], reflecting a situation that is reminiscent of high-income countries and links to the rapidly evolving food environment in these settings, and the increasing global burden of non-communicable diseases (NCD). Pregnant adolescent girls are a particularly vulnerable group because of the nutrient requirements of growth and development coupled with the heightened nutritional demands needed to support the fetus, and the sustained intergenerational effects of malnutrition [7,10]. Along with risks for the newborn, pregnancy in adolescence is associated with several negative health outcomes for the girl herself. Complications related to pregnancy and childbirth are the leading cause of death for girls aged 15–19 years [11]; a situation that is exacerbated by underlying nutritional disorders and chronic illness present in LMICs. With diet as a key risk factor for malnutrition, poor pregnancy outcomes, and NCD, it is clear that understanding the intake and practices of this group will be essential to promote better health and nutrition for adolescents and their future children.

To date, broad evidence syntheses on this topic have been limited and focused mainly on discrete populations of adolescents in high-income settings. A single review published in 2014 examined the dietary intake of schoolchildren and adolescents in LMICs and reported important findings consistent with the nutrition transition and the emerging double burden of malnutrition [12]. However, this study was descriptive in nature and combined data for boys and girls and schoolchildren (aged 6–10 years) and adolescents, making our review the first to look principally at the dietary patterns of adolescent girls in a quantitative manner that would allow for more informed policy and programming initiatives targeted towards this vulnerable population.

This review will summarize: (i) The current dietary intakes (types of foods consumed, macronutrients, and energy intake); (ii) patterns (frequency of consumption); and (iii) practices (snacking and skipping meals) of adolescent girls (10–19 years) in LMICs.

## 2. Materials and Methods

### 2.1. Literature Search

To access all relevant articles, we used the PICO methodology to develop a search strategy based on medical subject headings (MeSH) and key words that were identified a priori. Both grey literature and databases were searched. Databases included Medline, Embase, CAB Abstracts, CINAHL, Cochrane Controlled Clinical Trials Register, 3ie Databases of Impact Evaluations, and WHO regional databases (WHOLIS). The search strategy is presented in Appendix A.

### 2.2. Study Selection and Data Abstraction

Studies were managed using Covidence, online software used to streamline systematic review processes. Titles and abstracts were screened independently, while full text screening and data abstraction were completed in duplicate. Discrepancies were resolved with discussion or with the help of a third reviewer. Study eligibility criteria are summarized in the Appendix A. Studies were eligible if they included data collected in or after 2007 on the dietary intake or practices of adolescent girls (defined as 10–19 years) living in LMICs, based on the World Bank classification at the time of the search. The year 2007 was used as a cut-off because we were interested only in recent reports (within the last decade) of dietary intake that would reflect the changing food environments of adolescents. Studies were excluded if they were not in English, if the population was considered unhealthy (e.g., chronic and genetic diseases, nutritional disorders, metabolic disorders), or if authors did not disaggregate outcome data by sex (given our focus on adolescent girls). All study designs were eligible, with the exception of experimental designs that did not include a standard of care or usual practice group. Related systematic reviews were excluded, but reference lists were hand searched for any additional studies. All data was extracted into a standardized, previously piloted data abstraction form in duplicate. Though we collected information on additional outcomes (Appendix A), outcomes of interest reported here include types of food consumed, energy and/or macronutrient intake, snacking prevalence, and meal skipping prevalence.

### 2.3. Quality Assessment

Quality assessment of individual studies was completed in duplicate. Studies were critically appraised according to a set of criteria based on study type, using the Cochrane risk of bias guidelines and the National Institute of Health quality assessment tools [13,14]. Discrepancies were resolved through discussion. For all study types, including studies of observational design, we considered the following domains (where applicable): Sequence generation, allocation sequence concealment, blinding, incomplete outcome data, selective outcome reporting, and other sources of bias.

### 2.4. Data Synthesis

For each outcome of interest, data were synthesized and results were presented in qualitative (descriptive) or quantitative (tables of weighted means and prevalence) format. All analyses were weighted by sample size, in order to account for varying study size. Where reporting of estimates was not uniform across studies, conversions were made in order to pool data (e.g., median and interquartile range (IQR) was converted to mean and standard deviation (SD)) [15]. Outcomes were disaggregated by region, based on World Bank classification, and by age (10–14 years versus 15–19 years). For reporting within this paper, we have used *N* to represent the number of studies and *n* to represent sample size. 

#### 2.4.1. Dietary Intake

Food items were categorized according to an adapted version of FANTA’s minimum dietary diversity guide for women of reproductive age [16], into: (1) Grains, white roots, tubers, and plantains; (2) pulses (beans, peas, lentils); (3) dairy; (4) meat, poultry, and fish; (5) fruits; (6) vegetables; (7) snack foods; and (8) sugar-sweetened beverages (SSBs). Categories for nuts and seeds, eggs, oils and fats, condiments and seasonings, green leafy vegetables, vitamin A-rich fruits and vegetables, and other beverages were omitted due to data limitations. In order to capture the important variation in snack foods and to better distinguish snack foods from snacking as a meal pattern, this category was re-named and further sub-divided into: (7a) Sweet food items (including confectionary) and (7b) salty and fried food items [17]. We also added a category specific to fast foods. A food was considered a fast food if it was defined as such by the study authors. Common examples included burgers (hamburger, chicken burger), fried chicken, French fries, and pizza. Additional terms that authors may have used to describe fast foods were junk foods or high-fat foods. Occasionally we came across an unfamiliar food item, such as a traditional dish. In this instance, a Google search was performed that included the name of the food or meal and the country where the study was conducted. We then used the resulting information to appropriately classify these foods within our adapted FANTA guide. Where it was stated that a food item was consumed with another food item, the proportion was applied equally to each corresponding food group. For example, if 50% of girls from one study reported consuming lentils with rice, then we would report in our data abstraction sheet that pulses and grains were each consumed by 50% of girls sampled.

#### 2.4.2. Adequate Consumption of Fruits and Vegetables

The adequacy of fruits and vegetables was determined for adolescent girls using WHO recommendations [18]. These are population guidelines which recommend a daily intake of 2 servings of fruit and 3 servings of vegetables of 400 g/day (5 servings per day of 80 g each) [18]. To be included in this analysis, the study must have reported daily intake of fruits and vegetables, along with serving size data (i.e., g/day). Where studies reported fruit and vegetable consumption without associated serving size data, we could not determine adequacy.

#### 2.4.3. Macronutrient and Energy Intake

For our macronutrient analysis, we only included studies where data for all three macronutrients (carbohydrate, protein, and fat) were provided or where energy intake was reported along with any two macronutrients. Data on the final micronutrient could then be calculated to determine daily intake. Any implausible values were excluded from analysis. Data on macronutrient and energy intake were reported as grams per day and as kcal per day, respectively. Data were presented by geographical world region, and also pooled by urban and rural residence.

Data on energy and macronutrient intake were classified as being adequate or inadequate based on Institute of Medicine guidelines [19]. The recommended dietary allowance (RDA) for carbohydrates for adolescent girls is 130 g/day [19]. The RDA for protein is 46 g/day (for females 14–19 years) [19]. Currently, there is no adequate intake (AI), estimated average requirement (EAR), or RDA utilized for fat intake. However, the acceptable macronutrient distribution range (AMDR) for adolescents 10–18 years is 25–35% of total energy coming from fat [19].

#### 2.4.4. Meal Patterns

Breakfast skipping was defined as anything other than daily consumption of breakfast. We defined snacking as eating between meals, regardless of the time of day when the food was consumed. For our analysis, studies where authors have defined snacking based on the type of food being consumed (e.g., chips, biscuits, fruit, etc.) were included in the food group analysis rather than as a meal pattern. To be included, data must have indicated that foods were consumed between meals or as a meal replacement.

## 3. Results

We used a comprehensive conceptual framework (Appendix A), jointly developed by review authors and a technical advisory group comprised of members of the United States Agency for International Development (USAID)-funded Strengthening Partnerships, Results, and Innovations in Nutrition Globally (SPRING) project, to understand the determinants of adolescent girls’ dietary intake and eating practices as well as the resulting health outcomes.

From a total of 72,514 citations found through our database search, we screened 4455 full text articles and identified 288 studies for inclusion (Figure 1) after application of our specified inclusion and exclusion criteria. An additional 338 studies met our criteria, but were excluded because they did not disaggregate outcome data by sex. Several studies were excluded from analysis (*N* = 61) because of incomplete data (e.g., lacking sample size or SD) or data that was in a form that was unable to be pooled based on our outcomes of interest (e.g., dietary diversity scores).

Summarized study characteristics are summarized in Figure 2 and Table 1. As shown in the global map (Figure 2), several countries were overrepresented within our sample, including India (*N* = 43), Iran (*N* = 28), China (*N* = 25), and Brazil (*N* = 25), while countries from Europe and Central Asia were vastly underrepresented. Most studies were conducted among urban populations (59%), within school settings (78%), and were cross-sectional in design (67%), underscoring the need to apply caution when interpreting results in the context of external validity. The majority of included studies (64%) were deemed low quality, as they were small-scale cross-sectional studies that had high risk of selection bias due to purposive sampling, detection bias due to validity of data collection methods, and sometimes reporting bias due to incomplete data collection.

### 3.1. Energy and Macronutrient Intake

Data were available from 42 studies (*n* = 14,721) for analysis of energy and macronutrient (carbohydrate, protein, and fat) intake among adolescent girls. The weighted mean ± SD of energy intake among adolescent girls from all regions was 1840 ± 459 kcal/day. Mean daily energy intake was lower in South Asia (1411 ± 344 kcal/day) and Africa (1443 ± 177 kcal/day), when compared to all other regions (Figure 3). Among the same sample of adolescent girls, mean ± SD carbohydrate, protein, and fat intakes were 254 ± 70, 64 ± 20, and 59 ± 20 g/d, respectively (Figure 4). Similar regional patterns to energy intake were noted, whereby macronutrient intake of adolescents was significantly lower in Africa and South Asia, particularly for protein and fat (Figure 4).

We also disaggregated energy and macronutrient intake data by urban and rural residence. Mean energy intake was lower in rural settings (1621 ± 312 kcal/day) compared to urban settings (1906 ± 507 kcal/day). Similarly, mean fat and protein intake were lower among rural adolescents (35 ± 3 g/day and 49 ± 4 g/day, respectively) compared to those residing in urban settings (62 ± 21 g/day and 65 ± 20 g/day, respectively). Carbohydrate intake was similar across all residential settings.

### 3.2. Dietary Intake

We sought to determine the types of foods that adolescent girls in LMICs are consuming, and herein report daily consumption of each food group (Figure 5a). Intake for each food group disaggregated by region can be found in the Appendix A. Our analysis showed that 76% of all girls reported daily consumption of grains and, similarly, 76% reported consuming pulses and legumes daily. Daily consumption of dairy products was reported by only 16% of girls. Nearly half (46%) of girls sampled ate meat, poultry, and fish items, and fruits and vegetables were consumed daily by 44% and 37% of adolescent girls, respectively. A total of 4% of adolescent girls reported eating sweet snacks, 8% consumed salty and/or fried food items, 20% consumed fast foods, and 4% consumed SSBs on a daily basis.

Fast foods, salty and sweet snacks, along with SSBs are known to be common culprits of foods containing high amounts of fat, salt, and sugar. As energy-dense and nutrient-poor foods become more readily available in LMICs, and often appeal to school-aged children and adolescents, we aimed to evaluate how frequently adolescent girls consume these types of foods. Altogether 88 studies (sweet snacks: 18; salty snacks: 9; fast foods: 31; SSBs: 30) were included in our analysis of frequency of consumption of energy-dense foods. As seen in Figure 5b, the daily intake of these foods remained fairly low with the exception of fast foods, which was reported to be consumed daily by nearly 20% of adolescent girls in LMICs. 32% of girls consumed fast food items two to three times a week, and almost a quarter reported consuming these items four to six times a week. Overall, these energy-dense foods had the highest reported consumption at four to six times per week (sweet snacks by 63%, salty snacks by 78%, fast food by 23%, and SSBs by 49% of girls sampled).

### 3.3. Adequate Fruit and Vegetable Consumption

25 studies (*n* = 119,938) reported on the daily intake and serving size of fruits and vegetables, allowing us to determine adequacy based on WHO guidelines. The inadequate intake of both fruits and vegetables was widespread, as shown in Figure 6. Inadequate fruit and vegetable intake was highest in South Asia, with 97% (*n* = 850) and 90% (*n* = 14,478) of girls not consuming the recommended daily servings of fruits and vegetables, respectively. Compared to all other regions, inadequate fruit intake was lowest (32%) (*n* = 647) among girls in the Middle East and North Africa, while inadequate daily vegetable intake was lowest (48%) (*n* = 9992) among girls in the East Asia and Pacific region.

### 3.4. Meal Patterns

42 studies (*n* = 44,990) had data on the prevalence of breakfast skipping among adolescent girls. For all adolescents (10–19 years), 40.3% reported skipping breakfast (Figure 7a). It was slightly more common among adolescents aged 15–19 years (49%) compared to those aged 10–14 years (40%) (Figure 7a). We noted some regional differences (Figure 7b), with almost half of girls sampled in Africa and less than 20% of girls residing in Latin America and the Caribbean reporting that they skipped breakfast frequently.

We also looked at the proportion of adolescent girls who snack, defined as consuming food or drinks between meals. Of the studies that reported on snacking (*N* = 24; *n* = 12,647), it was found that 48.5% of adolescents (*n* = 6134) regularly consumed snacks throughout the day, with 55% of older adolescents snacking compared to 33% of younger adolescents. We also found that snacking was more common in the morning, between breakfast and lunch, and in the afternoon, between lunch and dinner, as compared to the evening.

## 4. Discussion

The results of this study have provided insight into the dietary patterns and habits of adolescent girls in LMICs, though results should be interpreted with caution given the non-representative nature of the studies included in this review. Nonetheless, our findings suggest a dietary transition that is reminiscent of “Westernized” diets (i.e., the nutrition transition [20]). Less than half of girls sampled reported eating dairy, meats (including poultry and fish), fruits, and vegetables and, even among those who consumed fruits and vegetables daily, servings were insufficient to meet WHO dietary guidelines. In contrast, we found high consumption of foods that have elevated fat, sugar, and salt contents such as fast foods, sweet and salty snacks, and SSBs. 20% of girls reported eating fast foods daily, while salty snacks and sweet snacks were consumed by 80% and 65% of girls, respectively, four to six times per week. Important regional variations were noted, with 40% of girls in Latin America and the Caribbean and 25% in South Asia reporting daily consumption of fast foods.

Carbohydrate intake was similar across all geographical regions and across urban and rural residence, while urban populations had higher intakes of fat and protein than did rural populations. Despite these differences, we found that mean macronutrient intake values were within the IOM guidelines [19]. Given that high carbohydrate intakes may be resulting, in part, from the increased consumption of processed snack foods, it is plausible that these food types are displacing the consumption of foods with adequate fat and protein, especially in rural areas. To support this theory, a recent study in Malaysian adolescents found that those attending rural schools had higher energy and cholesterol intakes compared to adolescents from urban schools, and that sugar and fat consumption was higher among obese adolescents in rural compared to urban settings [21]. Additionally, pooled data from 53 countries showed that the prevalence of overweight adolescents in rural areas is increasing globally, with the exception of Eastern Europe and Central and South Asia [22]. In North Africa, the prevalence of overweight adolescent girls was higher in rural (41%) as compared to urban areas (36%) [22]. Further, among Mexican adolescents, food insecurity was shown to have a positive association with consumption of refined grains and a negative association with fruit, vegetable, dairy, and high-protein food intake, highlighting the complex interaction between socio-demographic factors [23], dietary intake, and quality.

Others have shown similar findings that further highlight the nutrition transition and the resulting double burden of malnutrition in LMICs. A 2014 descriptive review in LMICs noted that the increased consumption of processed and fast foods among school-age children and adolescents, particularly those in urban settings, was likely to be contributing to the rising burden of overweight and obesity [12]. The convenient and low-cost nature of these foods, along with restaurants that provide a social atmosphere appealing to school-age children and adolescents, may be important factors contributing to their high intake. Authors also note that many diets are mostly plant-based (i.e., cereals, roots, and tuber), with low intake of fruits, vegetables, milk, and animal-source foods [12]. This lack of dietary diversity contributes to the additional burden of micronutrient deficiencies among this population. A more recent cross-cohort comparison of younger and older adolescents in Ethiopia, India, Vietnam, and Peru demonstrated changes in dietary intake consistent with the nutrition transition, including the high consumption of processed foods, sugary foods, and animal-source foods at the expense of plant-based foods [24]. Authors also note the convergence of adolescents’ dietary diversity across population sub-groups, including wealth brackets and urban and rural residence, which potentially highlights food environment factors, such as food promotion and advertising, that influence adolescent behaviors uniformly [24].

The 2016 Global Burden of Disease (GBD), Injuries and Risk Factor study has clearly demonstrated the negative repercussions of such “Westernized” diets, indicating that low fruit consumption, low whole grain intake, and high sodium were among the most important contributors to the 11.2% of diet-attributable deaths between 2006 and 2016 [25]. These deaths have increased in recent years, and diets high in red meat, SSBs, and low in milk have specifically been implicated in this surge [25]. Similarly, the PURE (Prospective Urban Rural Epidemiology) study reported lower risk of total mortality and non-cardiovascular mortality with higher intakes of fruit, vegetables (particularly raw vegetables), and legumes. Maximal benefits were reported for three to four servings per day (375–500 g/day) [26]. Given our findings, which suggest that the overall dietary habits of adolescent girls in LMICs are poor, coupled with results from the GBD work, the importance of improving adolescent nutrition is quite evident. However, nutrition interventions to improve dietary intake in these settings are lacking, despite some existing guidelines [27] and evidence that certain platforms, such as school and community-based services, may be beneficial [10]. Well-designed and long-term randomized controlled trials will be necessary to determine efficacy of interventions aimed at improving dietary intake and practices within this population.

Along with intake, we have highlighted dietary practices that could have implications for the health and wellbeing of adolescents. Snacking can be viewed as positive or negative depending on the type of snack foods being consumed (whether nutrient-dense or energy-dense) and, unfortunately, we were not able to make this distinction within our data. However, it is widely agreed that skipping breakfast can have detrimental consequences. Two systematic reviews looking at breakfast consumption and body weight outcomes found that the prevalence and risk of overweight was lower in adolescents who consume breakfast regularly compared to those who eat breakfast infrequently [28,29]. Breakfast skipping is also frequently associated with poor productivity and cognitive performance; this was demonstrated in a systematic review that also found more beneficial effects for children with poor baseline nutritional status [30]. More research should be focused on better understanding breakfast composition and its effects on health outcomes of adolescents, along with how to improve consumption of this meal for the 40% of adolescent girls who frequently skip it.

While this review was comprehensive in nature, it had several limitations. The analyses provided were limited by the information available and were neither nationally representative nor measured using consistent methods and indicators. As previously stated, several countries were overrepresented and the Europe and Central Asia region was especially underrepresented in our analyses. Additionally, data collection methods of primary studies varied and, as such, we pooled results from 24-h recalls, food frequency questionnaires, food records, and other methods. A sensitivity analysis was performed to determine whether this could have impacted results by re-analyzing consumption data by method and, while it did not change any interpretation, the various methods used to report intake made it challenging to pool data in a meaningful manner. 54% of studies that were included following title and abstract screening were found to not disaggregate data by sex, indicating substantial loss of available data for adolescent girls. Moving forward, including boys in discussions pertaining to dietary intake will be critical. Similarly, almost 80% of the studies were undertaken in a school setting, limiting their external validity considering a relatively high proportion of adolescents in LMICs are not in school. Lastly, standardization of guidelines—those that are specific to adolescent dietary intake (FANTA guidelines are geared towards women of reproductive age) and those that provide clarity on how to categorize energy-dense foods that are not currently found within dietary diversity guides—would be beneficial for future studies of consumption.

Given these limitations, one of the major take-home messages emanating from this review is the lack of good-quality, nationally-representative data on which to base recommendations for adolescent nutrition in LMICs. The absence of available dietary intake data has also been noted by USAID in their recent report on adolescent nutrition, which utilized DHS data from 2000–2017 [31]. Following discussion of results at an international consultation of nutrition experts in late 2017 (Stakeholders Consultation on Adolescent Girls’ Nutrition: Evidence, Guidance, and Gaps [32]), partners at SPRING, with the support of more than 100 academic, research, and non-governmental organizations, created a global call to action [33]. The aim of this call to action is to emphasize the importance of collaborative, international work that will generate the high-quality data needed to inform priorities for adolescent nutrition. Seven actions were cited, two of which highlight: i) The necessary development of standardized indicators for assessing adolescent health, nutrition, and wellbeing; and ii) proper sampling of this sub-group, disaggregation of programmatic data, and the inclusion of adolescents in national nutrition surveillance [33]. To move forward in this field will require consideration of each of these priority actions.

## 5. Conclusions

In conclusion, this review underscores the visible shift in dietary intake among adolescent girls in low- and middle-income settings, whereby “Westernized” diets may be gradually replacing more traditional diets. This shift will have major implications for the double burden of malnutrition that currently exists and will contribute to the rising prevalence of NCD. Moving forward, we will require evidence-based, impactful nutrition interventions that can help to combat this. However, the main message to be taken from this review underscores the need for better, more consistently measured, and representative data. As such, we recommend the development of standardized indicators and measurement tools that can be used to benchmark and track progress. Given the importance of good nutrition for adolescent girls, especially those who become pregnant, there is a critical and urgent need to address the current data limitations, such that future nutrition-related policies and programs would be better informed.

## Figures and Tables

**Figure 1 nutrients-10-01978-f001:**
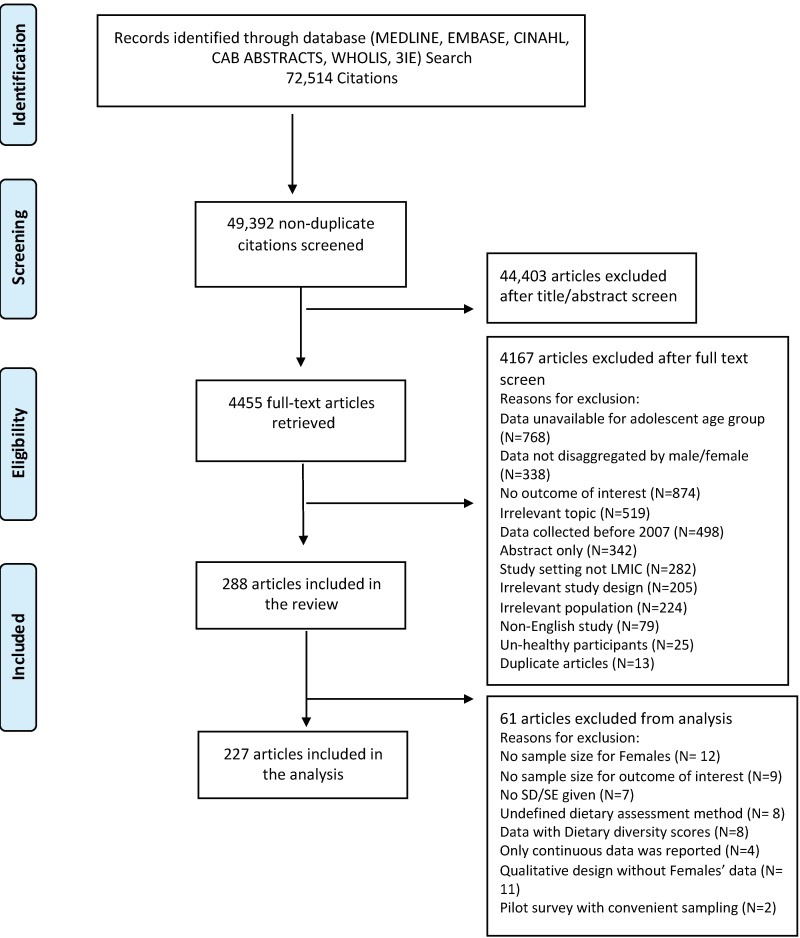
Flow diagram for study retrieval and selection.

**Figure 2 nutrients-10-01978-f002:**
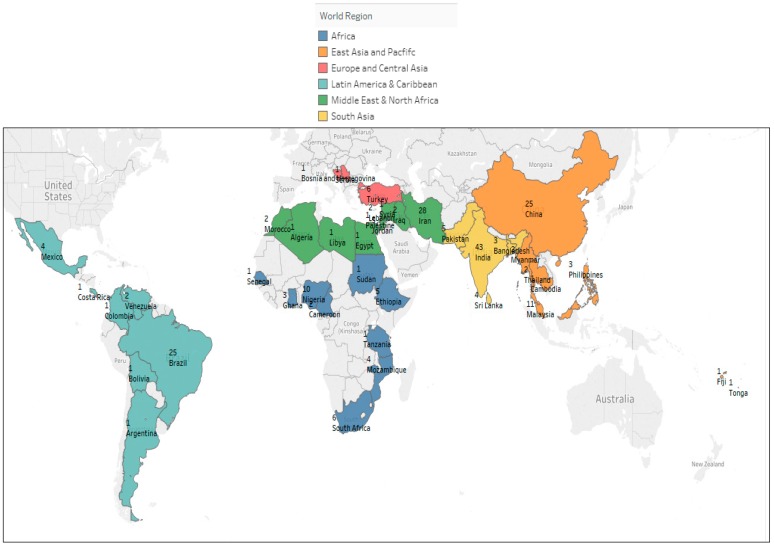
Global map representing records included by region.

**Figure 3 nutrients-10-01978-f003:**
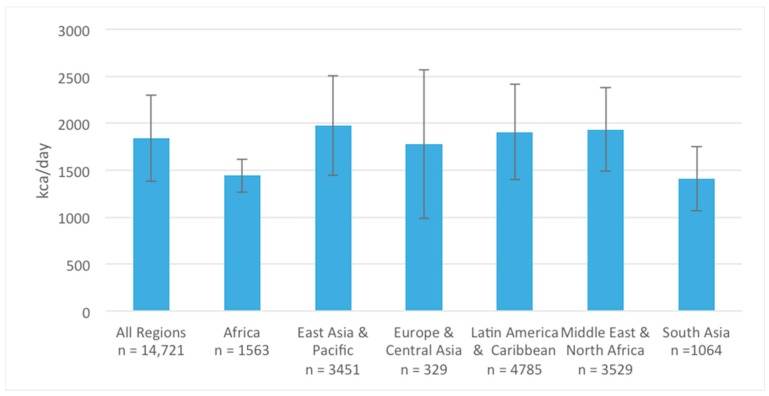
Adolescent girls’ energy intake (kcal/day) across different geographical world regions.

**Figure 4 nutrients-10-01978-f004:**
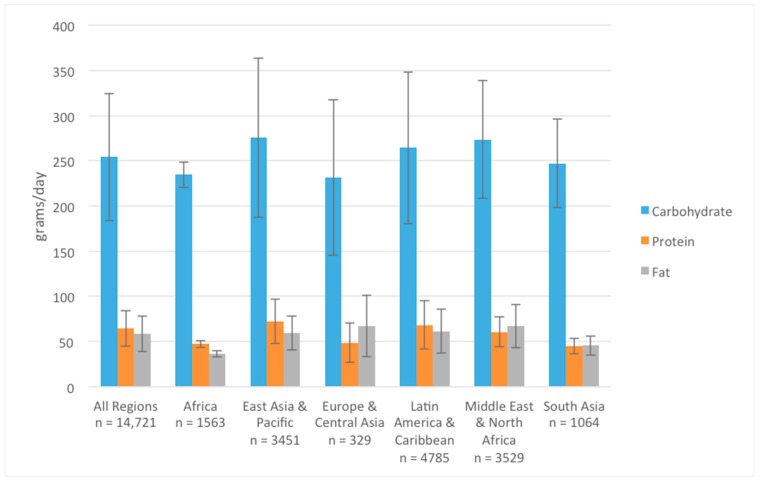
Adolescent girls’ macronutrient intake (g/day) across different geographical world regions.

**Figure 5 nutrients-10-01978-f005:**
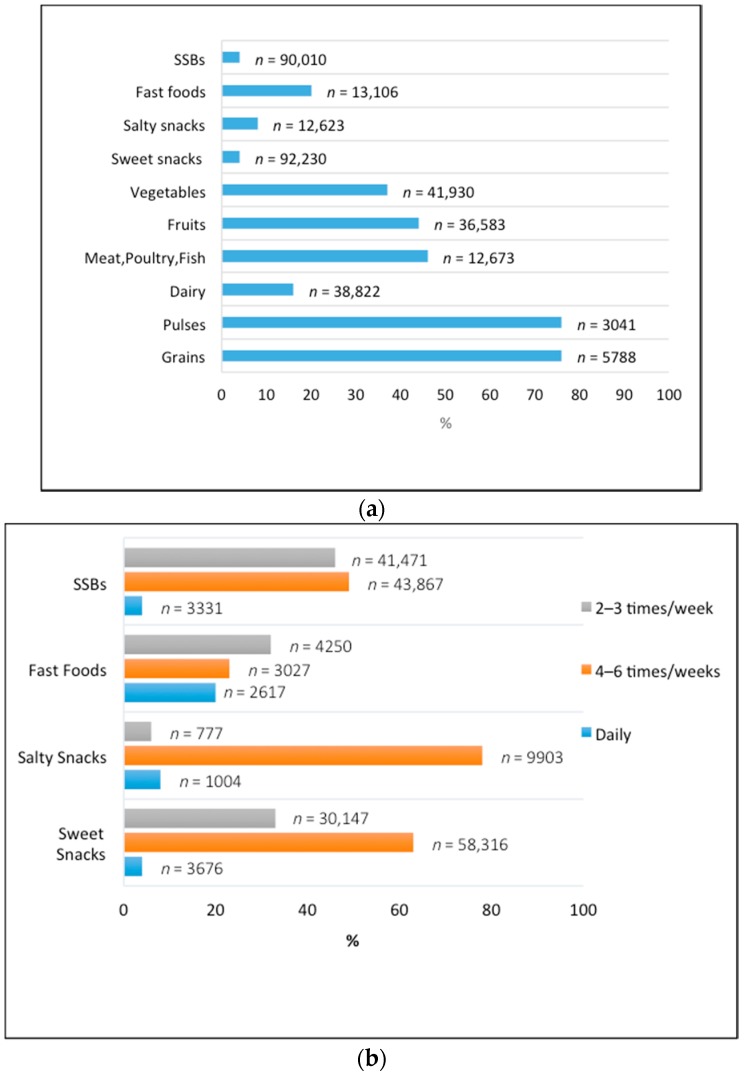
(**a**) Daily intake of food groups reported by adolescent girls; (**b**) Frequency of consumption of energy-dense foods (sweet and salty snacks, fast foods, and SSBs) by adolescent girls.

**Figure 6 nutrients-10-01978-f006:**
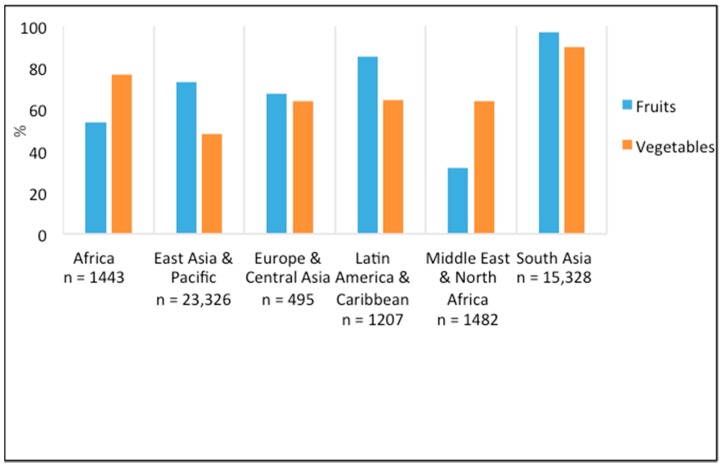
Proportion of inadequate fruit and vegetable intake (g/day) among adolescent girls across different geographical world regions.

**Figure 7 nutrients-10-01978-f007:**
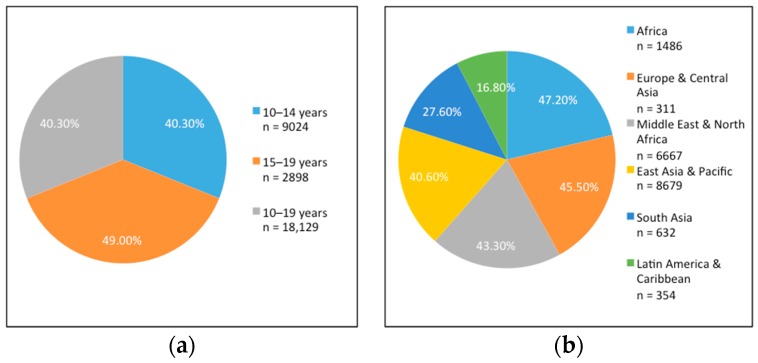
(**a**) Proportion of breakfast skipping practice among adolescent girls by age group. (**b**) Proportion of breakfast skipping practice among adolescent girls by region.

**Table 1 nutrients-10-01978-t001:** Summarized study characteristics.

Study Demographics			
World Regions	Study Design	Urban/Rural Setting	Study Setting
Africa (*N* = 36)	Cohort: *N* = 3 Cross Sectional: *N* = 33	Urban: *N* = 16 Rural: *N* = 6 Mixed: *N* = 11 Peri-Urban: *N* = 1 NR: *N* = 2	Community: *N* = 7 Regional: *N* = 4 School: *N* = 25
East Asia and Pacific (*N* = 47)	Cohort: *N* = 5 Cross Sectional: *N* = 39 Mixed Design: *N* = 1 RCT: *N* = 1 Twin Study: *N* = 1	Urban: *N*= 24 Rural: *N* = 6 Mixed: *N* = 14 Peri-Urban: *N* = 1 NR: *N* = 2	Community: *N* = 2 National: *N* = 4 Regional: *N* = 4 School: *N* = 37
Europe and Central Asia (*N* = 9)	Cross Sectional: *N* = 9	Urban: *N* = 8 Mixed: *N* = 1	Community: *N* = 1 School: *N* = 8
Latin America and the Caribbean (*N* = 35)	Case Study: *N* = 1 Cohort: *N* = 2 Cross Sectional: *N* = 26 Longitudinal: *N* = 1 Qualitative: *N* = 2 RCT: *N* = 3	Urban: *N* = 18 Rural: *N* = 1 Mixed: *N* = 11 Peri-Urban: *N* = 2 NR: *N* = 3	Community: *N* = 2 National: *N* = 5 Regional: *N* = 4 School: *N* = 24
Middle East and North Africa (*N* = 46)	Cross Sectional: *N* = 42 Qualitative: *N* = 1 Quasi-Experimental: *N* = 3	Urban: *N* = 33 Rural: *N* = 1 Mixed: *N* = 10 NR: *N* = 2	National: *N* = 1 Regional: *N* = 2 School: *N* = 43
South Asia (*N* = 54)	Case Control: *N* = 2 Cross Sectional: *N* = 50 RCT: *N* = 2	Urban: *N* = 35 Rural: *N* = 7 Mixed: *N* = 8 NR: *N* = 4	Community: *N* = 10 Regional: *N* = 3 School: *N* = 41
All Regions (*N* = 227)	Cohort: *N*= 10 Case Study: *N* = 1 Case Control: *N* = 2 Cross Sectional: *N* = 199 Mixed Design: *N* = 1 Longitudinal: *N* = 1 Qualitative: *N* = 3 Quasi-Experimental: *N* = 3 RCT: *N* = 6 Twin Study: *N* = 1	Community: *N* = 22 National: *N* = 10 Regional: *N* = 17 School: *N* = 178	Community: *N* = 22 National: *N* = 10 Regional: *N* = 17 School: *N* = 178

NR: Not reported; RCT: Randomized controlled trial.

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
