# Peer review of "The Dietary Intake and Practices of Adolescent Girls in Low- and Middle-Income Countries: A Systematic Review"

_nutrients, 2018, doi:10.3390/nu10121978_

Reviewer 1 Report

p.p1 {margin: 0.0px 0.0px 0.0px 0.0px; font: 10.0px Helvetica}

Figure 5: Consider spelling out macronutrient names in the key. 

Lines 324 - 326: I appreciate that the authors called attention to the lack of quality data and echoed this concern in their conclusions, as well as the coverage of some efforts to improve dietary intake data. This is a strong paper despite the data limitations and the authors build their credibility throughout. I would have liked to see a concluding recommendation to others in the field. 

Thanks for the thorough descriptions of the review tools used for this work. I learned a thing or two from reading the methods. 

Author Response

The Dietary Intake and Practices of Adolescent Girls in Low and Middle Income Countries: A Systematic Review

Manuscript ID: nutrients-393314

REVIEWER/EDITOR   COMMENTS

RESPONSE

Reviewer #1

Figure 5: Consider spelling out   macronutrient names in the key. 

We   have corrected this.

Lines 324 - 326: I appreciate that the   authors called attention to the lack of quality data and echoed this concern   in their conclusions, as well as the coverage of some efforts to improve   dietary intake data. This is a strong paper despite the data limitations and   the authors build their credibility throughout. I would have liked to see a   concluding recommendation to others in the field. 

Thank   you for these comments. We have included two additional statements in the   conclusion section (5) that pertain to our recommendations for improving   dietary intake along with data quality in general.

Reviewer 2 Report

Reviewers report: This systematic review explored the dietary intakes of LMIC adolescent girls. This is an important study which will provide an important contribution to the literature. However, there are a number of revisions required prior to publication as outlined below:

 Abstract

·         Page 1, line 23: The abstract needs to be independent of the main manuscript and so results need to be clearer. Initially upon reading the results I was unaware if this was % of total energy intake or % meeting recommendations for the particular foods. It was only upon reading the rest of the manuscript that it was the average % of girls reporting consuming these foods. This needs to be made clearer in abstract.

·         Page 1, line 25: Please provide the % of girls skipping breakfast to accompany the statement “skipping breakfast was common”

Introduction

·         Generally, I feel the introduction could be improved by providing a greater rationale for why this review is needed?  Has there been any other research on this topic? If so, what did they find and what were the implications? If not, explicitly state this is the first review on this topic to highlight the novelty of your review.

·         I would also like to see more prevalence statistics in this group i.e. % of adolescents that are  overweight in LMIC, micronutrients deficiencies etc

·         Page 1, line 38: Which Sustainable Development Goal targets are you referring to? There are 17 goals, will reaching adolescents be critical in achieving all of these? Also please provide reference to these targets.

Methods:

·         For consistency and clarity I would advise adhering to the PRISMA guidelines for reporting of this systematic review

·          Has this review been registered i.e., Prospero. If so please indicate registration number.

·         Page 2, line 75: What is the justification for using the cut-off year as 2007?

Results

The conceptual framework appears to be out of place here and it is not referred to again in the whole manuscript.

Table 1: Some information appears to be cut off under the heading urban/rural.

Table 1: There is a star (*) next to East Asian & Pacific but there is no information in the foot note of the table to confirm what this means.

Table 1: I think it would benefit having an extra column with “All regions”

Page 9, line 224: For consistency with methods, refer to this as meal patterns as opposed to dietary patterns. 

Discussion

·         Page 10, line 245: As dietary guidelines vary across different countries. Which guidelines are you referring to?

·         Page 11, line 293: You mention the importance of improving adolescent nutrition is evident but can you elaborate on how this could potentially be done in this population? Has there been any interventions carried out to target dietary intake among adolescents in LMIC.

Minor errors

  Page 10, line 255: remove s in foods types

Author Response

The Dietary Intake and Practices of Adolescent Girls in Low and Middle Income Countries: A Systematic Review

Manuscript ID: nutrients-393314

REVIEWER/EDITOR   COMMENTS

RESPONSE

Reviewer   #2

Abstract

Page 1, line 23: The abstract needs to be independent   of the main manuscript and so results need to be clearer. Initially upon   reading the results I was unaware if this was % of total energy intake or %   meeting recommendations for the particular foods. It was only upon reading   the rest of the manuscript that it was the average % of girls reporting   consuming these foods. This needs to be made clearer in abstract.

Thank   you for this comment. We have revised the abstract, such that the results can   now be read and understood independently from the body of the manuscript.

Page 1, line 25: Please   provide the % of girls skipping breakfast to accompany the statement “skipping   breakfast was common”

We   have included the % of girls who reported skipping breakfast.

Introduction

Generally,   I feel the introduction could be improved by providing   a greater rationale for why this review is needed?  Has there been any other   research on this topic? If so, what did they find and what were the   implications? If not, explicitly state this is the first review on this topic   to highlight the novelty of your review.

Thank   you for this comment. We have expanded on the rationale (line 81/82) for this   study. To our knowledge, there has been one related review that focused on   the dietary intake of adolescents in low and middle-income countries, but   this review did not quantify their data, nor did it focus on the adolescent   girl population. We have referenced it and elaborated on why an additional   review was needed. 

  I   would also like to see more prevalence statistics in this group i.e. % of   adolescents that are overweight in LMIC, micronutrients deficiencies etc.

We   have now included the most recent global estimates for underweight,   overweight, and select micronutrient deficiencies for adolescent girls.

 Page   1, line 38: Which Sustainable Development Goal targets   are you referring to? There are 17 goals, will reaching adolescents be   critical in achieving all of these? Also please provide reference to these   targets.

Thank   you. We have updated the text to reflect your comment.

Methods

For   consistency and clarity I would advise adhering to the PRISMA guidelines for   reporting of this systematic review

Thank   you for this suggestion. We have reviewed the PRISMA checklist and amended   the text where applicable.

Has   this review been registered i.e., Prospero. If so please indicate   registration number.

Unfortunately   this review was not registered with Prospero.

Page 2, line 75: What is the   justification for using the cut-off year as 2007?

We   were interested only in recent reports of dietary intake (within the last   decade), which could support the nutrition transition. We have included a   line to specify this.

Results

The   conceptual framework appears to be out of place here and it is not referred   to again in the whole manuscript.

We   agree it should not be in the main manuscript, and have moved it to the   supplementary material. However, we do feel it is important to include for   readers who want to understand adolescents’ dietary intake in the wider   context of determinants and health outcomes.

Table 1: Some   information appears to be cut off under the heading urban/rural.

We   have adjusted the ‘Urban/Rural Setting’ heading in the revised table.

Table 1: There is a star (*) next to   East Asian & Pacific but there is no information in the foot note of the   table to confirm what this means.

Thank   you for pointing this out. This star was needed for a previous version of the   table, so we have removed it.

Table 1: I think it would benefit having an   extra column with “All regions”

Thank   you for this comment. We have added this column.

Page 9, line 224: For consistency with   methods, refer to this as meal patterns as opposed to dietary patterns. 

Thank   you, we have corrected this.

Discussion

Page 10, line 245: As   dietary guidelines vary across different countries. Which guidelines are you   referring to?

This   is referring to the WHO guidelines for adequate fruit and vegetable   consumption, which is outlined and referenced in the methods (section 2.4.2).   We have added the word ‘WHO’ for clarity here.

Page 11, line 293: You mention the importance   of improving adolescent nutrition is evident but can you elaborate   on how this could potentially be done in this population? Has there been any   interventions carried out to target dietary intake among adolescents in LMIC.

Thank   you for this comment. We have included some text on the benefits of certain   delivery platforms for adolescents, as well as the need for more RCTs of   dietary intake interventions that target this population specifically.

Minor Errors

Page 10, line 255: remove   s in foods types

Corrected.  
